# Thirty-Four-Year Record (1987–2021) of the Spatiotemporal Dynamics of Algal Blooms in Lake Dianchi from Multi-Source Remote Sensing Insights

Jinge Ma [1,2,†], Feng He [3,†], Tianci Qi [1,2], Zhe Sun [1,2], Ming Shen [1], Zhigang Cao [1], Di Meng [3], Hongtao Duan [1,4,5] and Juhua Luo [1,*]

1   Key Laboratory of Watershed Geographic Sciences, Nanjing Institute of Geography and Limnology, Chinese Academy of Sciences, Nanjing 210008, China
2   University of Chinese Academy of Sciences, Beijing 100049, China
3   Kunming Institute of Dianchi Plateau Lakes, Kunming 650228, China
4   Shaanxi Key Laboratory of Earth Surface System and Environment Carrying Capacity, Northwest University, Xi'an 710127, China
5   College of Urban and Environment, Northwest University, Xi'an 710127, China
*   Correspondence: jhluo@niglas.ac.cn; Tel.: +86-025-86882165
†   These authors contributed equally to this work.

**Abstract:** Lake Dianchi is one of the most eutrophic lakes in China. The decline in water quality and the occurrence of massive algal blooms pose a significant threat to the health and environmental safety of the water ecosystem, making Lake Dianchi a key concern for algal bloom management in China. Obtaining the spatiotemporal dynamics of algal blooms for the longest time possible is crucial to algal bloom management and future prediction. However, it is difficult to acquire a long-term record of algal blooms from a single sensor in order to cover a more extended period of eutrophication in the lake due to the limitation of the spatial and temporal resolution of the sensors. In this study, Landsat and Moderate-Resolution Imaging Spectroradiometer (MODIS) images were combined with the Floating Algae Index (FAI) to reconstruct a unified time series of bloom areas to analyze the algal bloom dynamics in Lake Dianchi in recent decades. Regarding the interannual variation, the bloom area showed an increasing trend from 1987 to 2021, with larger bloom areas in 1991–1992, 2000–2003, 2012–2013, and 2020–2021. In terms of seasonal characteristics, the bloom area was significantly more prominent in the rainy season compared with the dry season during the year. The spatial distribution of the bloom frequency showed a pattern of higher frequencies in the north and lower frequencies in the south. From 2000 to 2021, the initial bloom time and bloom duration showed a trend of delaying and then advancing and decreasing and then increasing, respectively. We analyzed the importance of long-term records of algal blooms and found that the percentage of rainy season images is an essential factor in reconstructing time series based on different sensors. In addition, the relationship between environmental factors and algal blooms was analyzed. The results show that wind speed and air temperature were the main meteorological factors controlling the interannual variation in algal blooms in Lake Dianchi. Water quality factors such as nutrients have less of an influence on the variation in algal blooms because the algal growth demand has been met. Environmental management measures taken by local governments have led to improvements in the lake's trophic state, and continued strengthening of environmental pollution control is expected to curb the algal blooms in Lake Dianchi. This study provides a long-term record of algal blooms in Lake Dianchi, which provides essential reference information for a comprehensive understanding of the development process of algal blooms in Lake Dianchi and its sustainable development.

**Keywords:** algal bloom; Landsat; MODIS; long-term record; Lake Dianchi; multi-source remote sensing

## 1. Introduction

Lake Dianchi, located on the Yunnan–Guizhou Plateau in China and in the capital of Yunnan Province, Kunming, has been listed as one of the "Three Lakes" along with Lake Taihu and Lake Chaohu due to the frequent outbreaks of algal blooms in recent years and is a crucial target of the Chinese government [1]. Lake Dianchi is the only water body in its watershed that receives industrial, agricultural, and urban wastewater [2]. With the pollution, the eutrophication and phytoplankton biomass in the lake have increased dramatically since the 1980s, and algal blooms have become increasingly severe [3]. In order to achieve water management goals and predict future changes in the lake's ecosystem, it is crucial to obtain a long-term record of algal blooms in lakes (e.g., a record of the variation in the temporal and spatial distribution) [4,5], especially in the context of global warming and severe eutrophication. Without continuous, long-term observation of algal blooms, it is difficult to elucidate the cause-and-effect relationships of changes in the aquatic environment [6]. The longer the record, the more accurate the understanding of the history and current state of Lake Dianchi, which may provide decision support for the management and long-term data for the prediction of the aquatic environment. Therefore, we need to obtain reliable trends of algal blooms and reconstruct the eutrophication process in the 1980s in order to properly manage Lake Dianchi's aquatic environment.

Many studies on algal blooms in Lake Dianchi are based on chlorophyll a (Chl-a) data from field sampling [7–9], which can produce significant spatial and temporal discontinuities and is costly. In addition, the in situ monitoring sites at Lake Dianchi could not provide continuous data until the 1990s, and it was difficult to acquire algal bloom information before that. Over the past few decades, advances in sensor technology and remote sensing algorithms have made possible the otherwise challenging task of long-term monitoring of algal blooms [10–16]. Remote sensing provides more reliable information on algal blooms than traditional methods [17].

Researchers have used such sensors as Landsat, the Moderate-Resolution Imaging Spectroradiometer (MODIS/Terra), the Ocean and Land Color Imager (OLCI), Gaofen-5 (GF-5), and HuanJing-1 (HJ-1) to monitor the dynamics of algal blooms in Lake Dianchi [2,18,19]. Zhao, et al. [20] used Landsat imagery to obtain the spatial and temporal dynamics of algal blooms in Lake Dianchi from 1986 to 2016; however, due to the limitation of Landsat's temporal resolution (16 days), it was not possible to monitor the rapidly changing characteristics of algal blooms [21]. Jing, Zhang, Hu, Chu and Ma [2] conducted high-frequency observations of algal blooms in Lake Dianchi from 2000 to 2018 using MODIS/Terra data (2000~). However, it was difficult to obtain information on algal blooms in the initial stage of eutrophication in Lake Dianchi (before 2000) due to the limitation of the sensor service years. Since the Coastal Zone Color Scanner (CZCS) currently has no valid data on Lake Dianchi, and the Sea-viewing Wide Field Sensor (SeaWiFS) was launched in 1997, Landsat data, as the longest-archived satellite data that are currently available, represent the best data for constructing and expanding a time series of the bloom area at Lake Dianchi.

Considering the lack of a long-term record of algal blooms in Lake Dianchi since the 1980s (the period of rapid eutrophication), this study aimed to achieve a reliable reconstruction of the spatial and temporal dynamics of the algal blooms that have occurred in Lake Dianchi since the 1980s using Landsat and MODIS/Terra images. In particular, we: (1) constructed algal bloom extraction methods based on Landsat and MODIS/Terra images and acquired a dataset on algal blooms in Lake Dianchi from 1987 to 2021; (2) compared the Landsat and MODIS/Terra algal bloom time series and constructed a dual-sensor long-term record of algal blooms in Lake Dianchi from the 1980s to the present; and (3) analyzed the spatial and temporal variation in algal blooms in Lake Dianchi since the 1980s and the driving forces of algal blooms with meteorological and water quality data. Our results may help us trace the spatial and temporal characteristics of algal blooms in Lake Dianchi since the period of its rapid eutrophication and provide valuable references for algal bloom control and prevention measures in Lake Dianchi and other lakes.

## 2. Materials and Methods

### 2.1. Study Area

Lake Dianchi (102°36′–103°40′E, 24°40′–25°02′N) is located in the middle of the Yunnan–Guizhou Plateau in Kunming City, the capital of Yunnan Province, in southwest China [22] (Figure 1). The average depth of Lake Dianchi is 5 m, and its maximum depth is 8 m. The surface area is about 310 km$^2$, and Lake Dianchi is the largest freshwater lake in Yunnan Province [8,23]. The lake is 1887.4 m above sea level, with a shoreline of 150 km. An artificial dam divides the lake into Caohai and Waihai (Figure 1a), with Caohai in the north covering only 7.83 km$^2$ and Waihai in the south covering 286.78 km$^2$, accounting for 97% of the total area [24]. The urbanized land (mainly the urban area of Kunming City), which accounts for 30.17% of the watershed, is concentrated in the eastern and northern parts of Lake Dianchi [25]. Several rivers in the watershed flow into the lake through Kunming City, discharging a significant amount of urban and agricultural sewage into Lake Dianchi (Figure 1b). With the economic and social development in the watershed, the water quality of the rivers in Lake Dianchi has been declining, and the water pollution problems in the lake have become more prominent, especially after the 1980s, when the water quality deteriorated at an accelerated rate [8]. The water quality of Lake Dianchi declined rapidly within a short period of time, the eutrophication of the lake increased, and algal blooms frequently occurred [23].

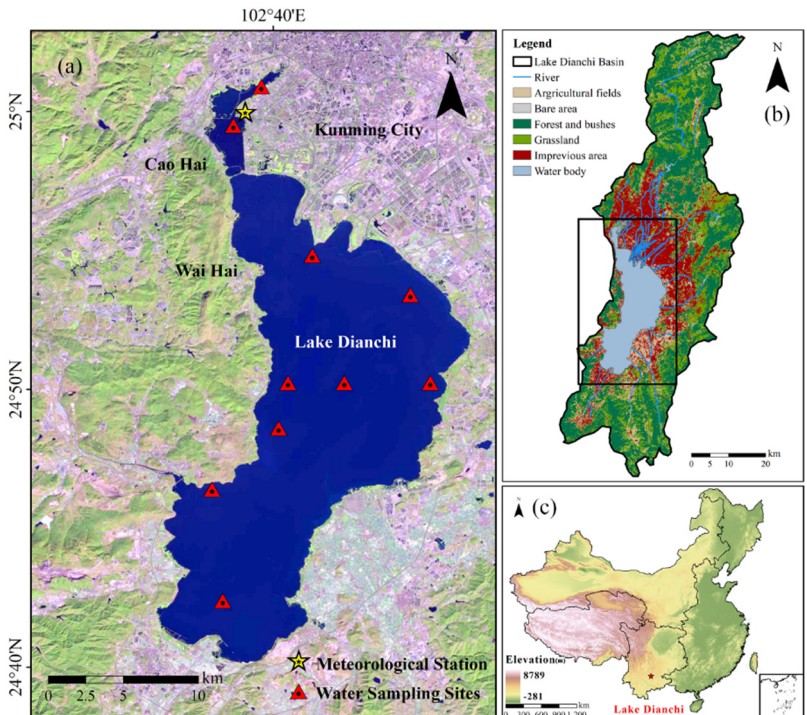

**Figure 1.** Distribution of sampling sites and rivers and the location of the Lake Dianchi Basin. (**a**) Distribution of sampling sites and the meteorological station in Kunming City. The yellow star marks the location of the national meteorological station. The red triangle marks the location of the sampling sites in Lake Dianchi. (**b**) Land use in the Lake Dianchi Basin. The black box indicates the location of Lake Dianchi. (**c**) Location of Lake Dianchi in China.

### 2.2. Satellite Data

#### 2.2.1. Landsat Data

USGS Landsat-5 TM/Landsat-7 ETM+/Landsat-8 OLI surface reflectance products were obtained from Google Earth Engine (https://earthengine.google.com, accessed on 20 December 2021) and used in this study. The spatial resolution of Landsat is 30 m with a 16-day observation period. The acquisition period was January 1987 to December 2021. After cloud masking (for details, see Section 2.3.2), images in which the remaining pixels

exceeded more than 50% of the lake were selected (for a total of 314 images, Figure 2a). These images contain a quality control band (QA Band) that provides information on such elements as clouds, shadows, snow and ice, and cirrus clouds. In May 2003, the Landsat-7 ETM+ scan line corrector (SLC) failed, resulting in a data strip loss that rendered the Landsat-7 ETM+ images unavailable. Therefore, all of the Landsat-7 ETM+ images used in this study are pre-2003 images.

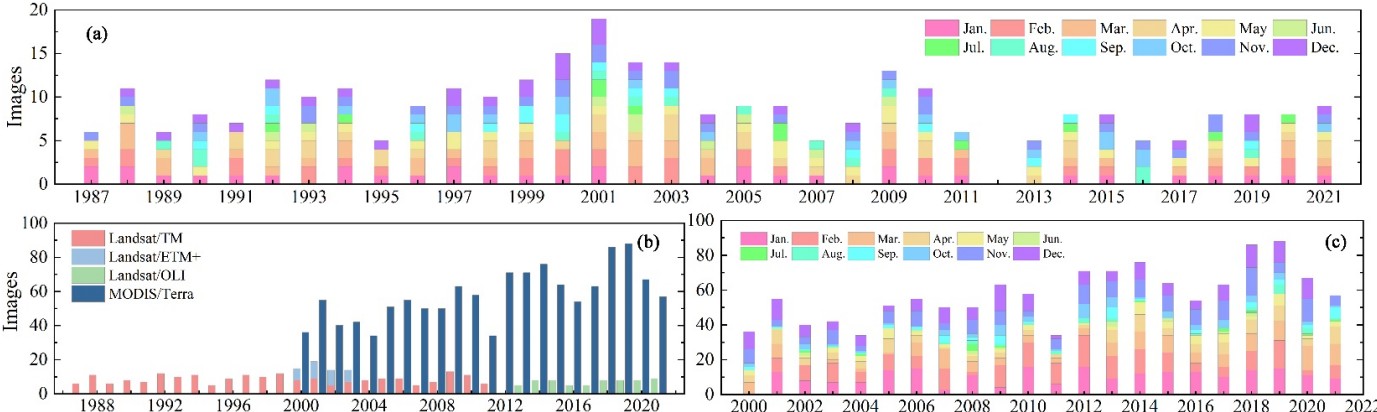

**Figure 2.** Temporal distributions of the Landsat TM/ETM+/OLI and MODIS/Terra scenes examined in this study. (**a**) Distributions of the Landsat scenes examined in this study. (**b**) Distributions of the Landsat TM/ETM+/OLI and MODIS/Terra scenes examined in this study from the period 2000–2021. (**c**) Distributions of the MODIS/Terra scenes examined in this study.

### 2.2.2. MODIS/Terra Data

MODIS/Terra Level-1A data on Lake Dianchi from February 2000 to December 2021 were obtained from NASA's OceanColor Web (https://oceandata.gsfc.nasa.gov/, accessed on 12 Febuary 2021). The Level-1A images were processed to generate Level-1B images using SeaDAS 7.5.1 (vicarious calibration file: R2018). Then, the Level-1B images were partially atmospherically corrected to remove the ozone and water vapor absorption and Rayleigh scattering of atmospheric molecules. The Raleigh-corrected reflectance ($R_{rc}$, dimensionless) was obtained as a reference [26]. Finally, the calculated $R_{rc}$ data were mapped to an equidistant cylindrical projection. The data at the 469 nm and 555 nm bands with an original resolution of 500 m were resampled to a resolution of 250 m. The $R_{rc}$ data at three bands (645, 555, and 469 nm) were used to generate 'true-color' composite images with a resolution of 250 m. A total of 1265 cloud-free MODIS/Terra images of the entire lake were acquired by visual inspection, excluding images containing clouds and poor-quality images of the lake (Figure 2c).

### 2.3. Algal Bloom Extraction
#### 2.3.1. Pre-Processing of Satellite Images

Cloud pixels have a high FAI, which makes it difficult to extract algal blooms accurately, so they need to be identified and removed during processing. For the Landsat-5 TM/Landsat-7 ETM+/Landsat-8 OLI images, cloud and cloud shadow pixels were identified using the QA band (pixel_qa). For the MODIS/Terra $R_{rc}$ images, since images showing the presence of thick aerosols or sunglint were removed during the visual inspection (for details, see Section 2.2.2), we only carried out an analysis to distinguish clean water and algal blooms from cloud pixels. There are several widely used de-clouding thresholds for inland waters, such as $R_{rc}(1640) = 0.0215$ [27] and $R_{rc}(1640) = 0.03$ [28]. We selected 137 images, manually checked the cloud (N = 35,737), algal bloom (N = 45,623), and clear water (N = 32,093) images, and then counted the numbers of pixels in the acquired images at different bands. The results show (Figure S1) that $R_{rc}(1640) = 0.0215$ and $R_{rc}(1640) = 0.03$ masked a large number of algal bloom pixels, and $R_{rc}(1640) = 0.0215$ even masked some clear water pixels. Therefore, none of these thresholds can be used to obtain good re-

sults on Lake Dianchi. So, we developed a new threshold to distinguish cloud pixels in MODIS/Terra $R_{rc}$ data. The results also show (Figure S1) that the $R_{rc}(2130)$ band can distinguish cloud pixels and clear water pixels well. The upper limit of 0.0246 for algal bloom pixels was used as the cloud mask threshold. We verified the accuracy of $R_{rc}(2130) = 0.0246$ as the cloud mask threshold with randomly selected samples (for details, see the Supplementary Material). The verification results (Table S1) show that the threshold achieves a high accuracy rate with an OA above 98% and a Kappa coefficient of 0.9011. In addition, based on a visual inspection (Figure S2), $R_{rc}(2130) = 0.0246$ was found to be able to mask most cloud pixels while retaining most algal bloom and clear water pixels. In addition, we shrunk the lake boundary inward by 2 pixels (about 500 m) in order to avoid false-positive results caused by the influence of land pixels.

### 2.3.2. Floating Algae Index

The Floating Algae Index (FAI) calculated in the red, near-infrared, and short-wave infrared bands was used to extract the areas in which algal bloom outbreaks occurred using thresholds. This method uses the uplift of algal blooms in the near-infrared band to emphasize the waters affected by algal blooms and reduce the FAI's sensitivity to different aerosol types by subtracting the baseline [16]. Due to its excellent performance, the FAI has been used in many studies [29,30]. The formula for the FAI is as follows:

$$FAI = R_{rc,NIR} - R'_{rc,NIR},$$
$$R'_{rc,NIR} = R_{rc,Red} + (R_{rc,SWIR} - R_{rc,Red}) \times (\lambda_{NIR} - \lambda_{Red})/(\lambda_{SWIR} - \lambda_{Red}) \tag{1}$$

where $R'_{rc,NIR}$ is the baseline $R_{rc}$ at the near-infrared band (555 nm) and $R_{rc,Red}$, $R_{rc,NIR}$, and $R_{rc,SWIR}$ are the $R_{rc}$ in the red, near-infrared, and short-wave infrared bands, respectively.

### 2.3.3. Thresholds for Distinguishing Algal Bloom Pixels

The maximum gradient method was used to obtain the extraction threshold of algal blooms. This method has been widely used to identify wetlands, waters, and mining vessels [28,31,32]. First, the FAI was calculated for each image, and then the corresponding gradient image was obtained. The pixels of clear water and dense algal blooms were removed using FAI < −0.01 and FAI > 0.02, respectively, in order to retain the pixels at the boundary of the bloom area [28]. The pixels that were retained ranged between pure water and thick algae scum pixels and were at the edge of the algal bloom range. If a scene has no algae at all, then all pixels in the scene are excluded by FAI < −0.01, and no response threshold is generated to influence the final threshold determination. Because the FAI of algal bloom waters is far greater than that of clear waters, the gradient in the FAI was the largest at the boundary between algal-bloom-infested and clear waters. The FAI corresponding to the pixels at the maximum spatial gradient was used as the threshold for determining the boundary between algal-bloom-infested and clear waters. To exclude outliers, a histogram was constructed using a set of pixels around the boundary rather than the gradient values of individual pixel points [31], and the results show an approximately normal distribution. The thresholds for all the images were statistically analyzed to ensure a that a consistent threshold would be maintained over the whole period. The mean of all thresholds minus two times the standard deviation was used as the threshold for identifying algal bloom pixels. The thresholds obtained by subtracting the mean value from the 2-sigma standard deviation were able to cover 95% of the images and avoid the false positives caused by the use of the minimum threshold in the statistics. The method was applied to the Landsat TM/ETM+/OLI and MODIS/Terra $R_{rc}$ images, and $T_{Landsat} = 0.01124$ and $T_{MODIS/Terra} = -0.00778$, respectively, were determined to be the thresholds for this study (Figure S3). In summary, the method used in this study excluded non-bloom pixels and a universal threshold for most images was obtained based on the statistical results of the approximately normal distribution. Moreover, we examined the consistency of the algal

bloom results obtained from different sensors by using MODIS/Terra and Landsat on the same day.

In this study, the bloom area rather than the bloom intensity was applied as an indicator to reveal the spatial and temporal variation in algal blooms in Lake Dianchi because the bloom area data were the only verifiable data we could use. Some studies have used bloom intensity as an evaluation indicator for algal blooms. Ho, et al. [33] used a single band (the near-infrared band) to characterize bloom intensity and applied it to the assessment of lakes around the world, but this work has been questioned by Feng, Dai, Hou, Xu, Liu and Zheng [21] and there is some doubt about the application of this method. Binding, et al. [34] used pigment concentration as a measure of bloom intensity. Wang, et al. [35] modeled the AFAI with measured biomass data and revealed the bloom intensity of Sargassum. However, all of these studies required a large amount of simultaneous satellite and in situ data. Although we obtained water quality data from automated stations and field surveys in Lake Dianchi, we lacked simultaneous data on biomass and pigment concentrations. Therefore, modeling and validation data on the bloom intensity were difficult to obtain for this study. From another perspective, the relationship between spectral indices (e.g., the FAI and AFAI) or pigment concentrations of algal blooms and bloom intensity remains unclear. The relationship between AFAI and algae biomass is not stable [35]. Moreover, the pigment concentration of the algal bloom area is difficult to accurately obtain by remote sensing [21]. We can conclude that the methods that are currently used to obtain the intensity of an algal bloom from algal bloom pixels remain immature and fall outside the scope of this study.

*2.4. Environmental Factors*

In order to analyze the relationship between algal blooms and environmental factors, meteorological and water quality data on Lake Dianchi were obtained. The average wind speed, average air temperature, maximum air temperature, minimum air temperature, 20–20 h precipitation (the precipitation from 20:00 to 20:00 on the following day), average air pressure, and number of sunshine hours at Kunming Station from 1987 to 2021 (No. 56,778, 102°23′24″E, 25°N, altitude of 1889.1 m) were obtained from the Chinese Meteorological Data Service Center (http://data.cma.cn/, accessed on 28 October 2021) for subsequent analysis. The water temperature, pH, ammonia nitrogen ($NH_3$-N), total phosphorus (TP), and total nitrogen (TN) at the national control sampling sites (red triangles in Figure 1a) at Lake Dianchi from 1987 to 2018 were obtained from the Kunming Environmental Monitoring Center. Details of the meteorological and water quality data used in this study are shown in Table 1.

**Table 1.** The minimum, median, mean, and maximum monthly values of physical, chemical, and biological variables in Lake Dianchi. Abbreviations: TN, total nitrogen; TP, total phosphorus.

|  | Mean | Minimum | Maximum | Median |
|---|---|---|---|---|
| Water Temperature (°C) | 17.96 | 7.20 | 28.70 | 18.00 |
| pH | 8.82 | 6.27 | 9.95 | 8.75 |
| NH3-N (mg/L) | 0.26 | 0.03 | 1.32 | 0.27 |
| TP (mg/L) | 0.16 | 0.03 | 3.28 | 0.16 |
| TN (mg/L) | 1.86 | 0.40 | 6.46 | 2.04 |
| TN/TP ratio (mass) | 16.46 | 4.31 | 59.35 | 13.16 |
| Precipitation (mm) | 81.68 | 0.00 | 474.90 | 45.50 |
| Air Pressure (hPa) | 810.60 | 805.30 | 816.50 | 810.27 |
| Wind Speed (m/s) | 2.15 | 0.80 | 4.40 | 2.10 |
| Air Temperature (°C) | 15.59 | 5.60 | 21.90 | 16.65 |
| Sunshine Hours (h) | 180.45 | 44.50 | 322.00 | 44.50 |

*2.5. Statistical Methods*

2.5.1. Analysis of Algal Bloom Time Series

The Landsat and MODIS/Terra time series of bloom areas between 2000 and 2021 were compared, and the most consistent time series were selected in order to analyze the algae dynamics that have occurred in Lake Dianchi since the 1980s. The time series of bloom areas obtained from Landsat and MODIS/Terra were obtained at monthly, one-year, two-year, three-year, and four-year intervals. A linear fit was performed for each time series from 2000 to 2021, and the slope difference between the Landsat and MODIS/Terra time series based on the same time scale was calculated. The time scale of the unified time series of algal blooms was determined based on the time scale with the smallest slope difference. It is worth noting that the results of Landsat were used before 1987–1999, and a combination of MODIS/Terra and Landsat was used after 2000 (if data from both MODIS/Terra and Landsat were available on the same day, the result with the larger bloom area was selected on that day).

In addition, the initial bloom time of a year was defined as the average day of the year (doy) on which the first three algal blooms (whose area is greater than 5%) occurred in that year. The end time was defined as the average doy of the last three algal blooms. The bloom duration time was defined as the end time minus the initial bloom time of the current year. We only analyzed the results after 2000 for annual indicators such as the bloom frequency, the bloom duration time, and the initial bloom time because of the small number of practical observations from Landsat.

2.5.2. Statistics of Environmental Factors

The collected data were processed in order to demonstrate the meteorological and water quality changes that occurred in Lake Dianchi during the study period. The month-by-month data obtained in this study were used to obtain the annual mean for the meteorological data. For the water quality data, the monthly mean from each sample site was used as the monthly mean of this factor for that month, and all monthly means for each year were used to obtain the annual mean. The algal bloom indicators (bloom area, initial bloom time, and bloom duration time) were used as the corresponding parameters, linear correlations were obtained, and the correlation and significance of *p*-values were calculated.

2.5.3. Statistical Metrics

The root square mean error (*RSME*), the mean absolute percentage error (*MAPE*), the relative error (*RE*), and the coefficient of determination (R$^2$) between the extracted bloom areas from Landsat data and MODIS/Terra data were used to analyze the consistency between Landsat and MODIS/Terra. The formulas for the RSME, MRE, and RE are as follows:

$$RSME = \sqrt{\frac{\sum_{i=1}^{N}(X_{L,i} - X_{M,i})^2}{n}} \tag{2}$$

$$MAPE = \frac{100\%}{n}\sum_{i=1}^{n}\left|\frac{X_{L,i} - X_{M,i}}{X_{L,i}}\right| \tag{3}$$

$$RE = (X_{L,i} - X_{M,i}) \times 100\% \tag{4}$$

In Equations (2)–(4), $X_L$ is the bloom area extracted from Landsat, $X_M$ is the bloom area extracted from MODIS/Terra, and *n* denotes the total number of samples.

**3. Results**

*3.1. Combining the Landsat and MODIS/Terra Observations*

The trends of the time series of the algal bloom area obtained by Landsat and MODIS/Terra at different time scales were analyzed (Figure 3). From the monthly scale, due to the insufficient number of Landsat observations, the characteristics of double peaks in the MODIS/Terra time series are not reflected in the Landsat time series. The slope

difference between the Landsat time series (yl = −1.9759x + 4012.0) and the MODIS/Terra time series (ym = 0.0326x − 3.1128) is the largest (Dif = 2.0085). In the time series of other scales, the slope difference (Dif = 1.6114) between the three-year Landsat time series (yl = −1.2948x + 2640) and the MODIS/Terra time series (ym = 0.185x − 180.56) is larger than the one-year time series (Dif = 1.4068). The slope difference between the Landsat time series (yl = −1.2023x + 2452.8) and the MODIS/Terra time series (ym = −0.0402x + 123.3) is 1.1621, which is the smallest slope difference. As the two-year time series can represent more information, the Lake Dianchi time series before 2000 was constructed on a two-year scale.

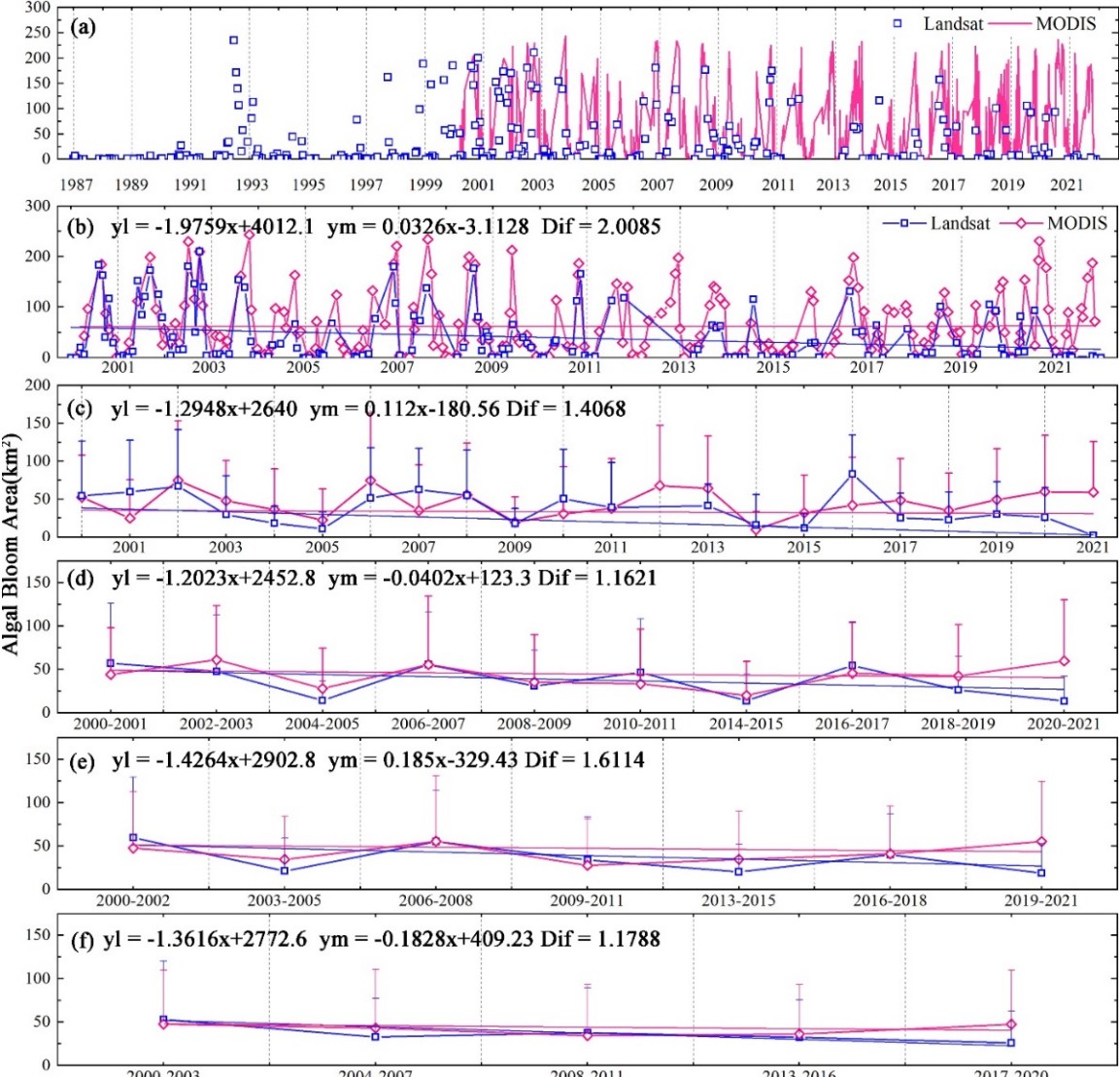

**Figure 3.** Time series of different time scales of the average algal bloom area in Lake Dianchi between the Landsat and MODIS/Terra results (during 2000–2021). The time intervals of (**a–f**) are 1 day, 1 month, 1 year, 2 years, 3 years, and 4 years, respectively. "ym" represents the fitting equation of the time series obtained from MODIS/Terra images at the current time scale. "yl" represents the fitting equation of the time series obtained from Landsat images at the current time scale. "x" represents the independent variable of the fitting equation of the time series at the current time scale. "Dif" represents the slope difference between the MODIS/Terra time series and the Landsat time series. The lower the Dif, the greater the similarity of the trends between the time series.

### 3.2. Long-Term Records of Algal Blooms since the 1980s

From 1987 to 2021 (Figure 4a), the average area of algal blooms in Lake Dianchi showed an upward trend (y = 1.3387x + 27.976, *p* = 0.174). From 1987 to 1999, the average

area of algal blooms in Lake Dianchi showed an upward trend (from 5.4 km$^2$ in 1987–1988 to 52.24 km$^2$ in 1997–1999), and the average area in 1991–1992 reached its peak value (67.2 km$^2$). After 2000, the algal blooms in Lake Dianchi deteriorated, and the average bloom area increased to 35.45 km$^2$ during 2000–2021 from 22.46 km$^2$ during 1987–1999. From 2000 to 2021, the average area of algal blooms in Lake Dianchi showed no significant upward or downward trend.

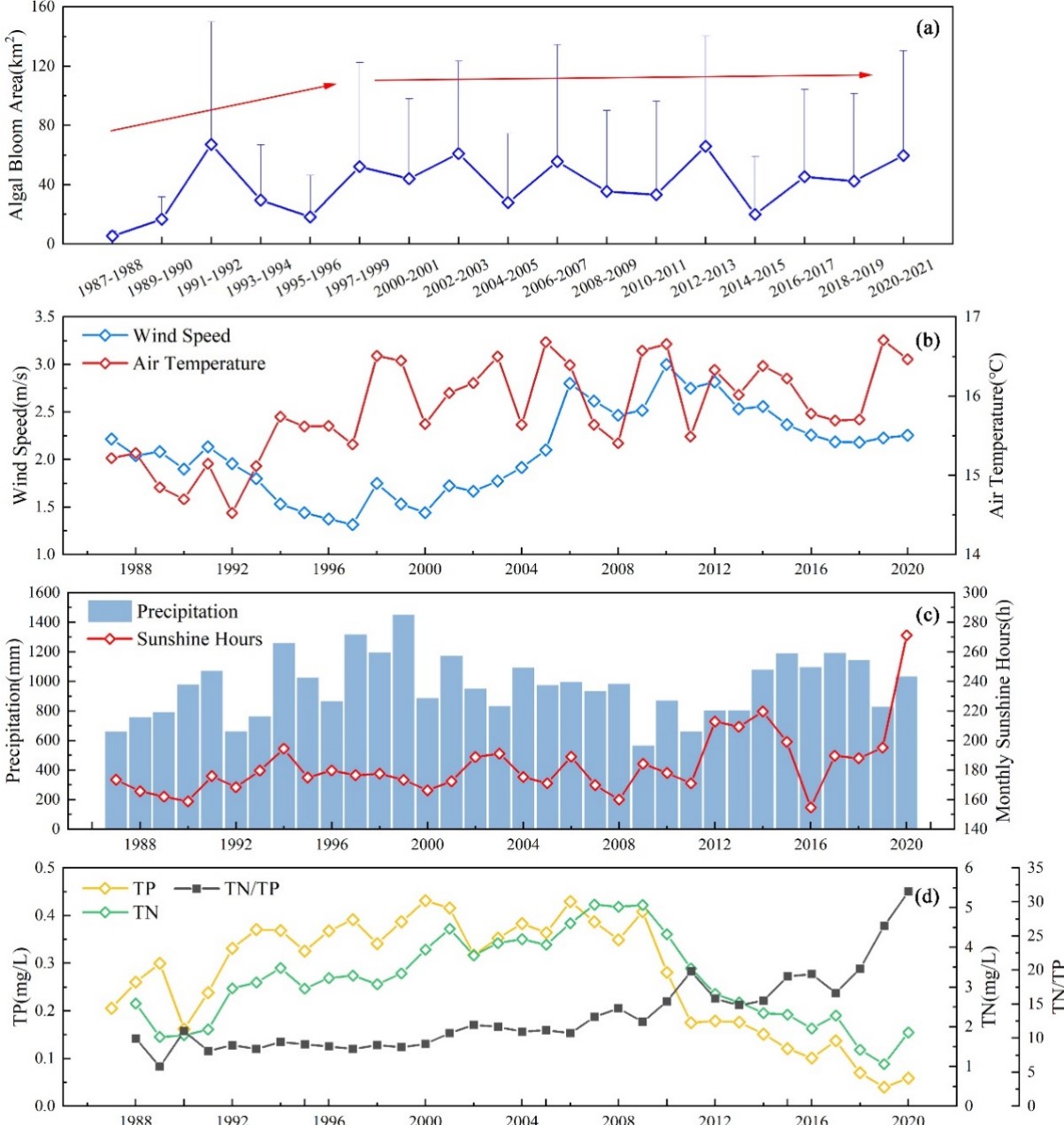

**Figure 4.** (**a**) Time series on a 2-year scale of the algal bloom area in Lake Dianchi between Landsat and MODIS/Terra (during 1987–2021). (**b**) Time series of the average wind speed and air temperature during 1987–2020. (**c**) Time series of the average precipitation and hours of sunshine during 1987–2020. (**d**) Time series of the average total nitrogen (TN), total phosphorus (TP), and N/P ratio (TN/TP) during 1987–2020.

As for the monthly bloom area in Lake Dianchi (Figure 5a), the algal blooms have noticeable cyclical changes every month, and the bloom area in the rainy season (May–October) is significantly larger than that in the dry season (November–April of the following

year). From January to March, the bloom area in Lake Dianchi was relatively small (it did not exceed 20 km$^2$), and the area was the smallest in February (only 6.02 km$^2$). With the increase in the air temperature, the bloom area increased to about 50 km$^2$ from April to May. From June, Lake Dianchi reached the peak period of algal blooms, and the average algal bloom area was greater than 80 km$^2$ until November, exceeding 25% of the entire lake area. The average bloom area was the highest in July from June to November, reaching 128.02 km$^2$, accounting for 42.9% of the lake area. The average bloom area from August to October exceeded 100 km$^2$. It can be seen that the bloom area in August (107.19 km$^2$) was lower than that in July (128.02 km$^2$) and September (121.97 km$^2$), which may have been caused by the influence of clouds and rain in August and the smaller number of solid images. In December, the bloom area decreased to 50.23 km$^2$ and reached its lowest value the following February.

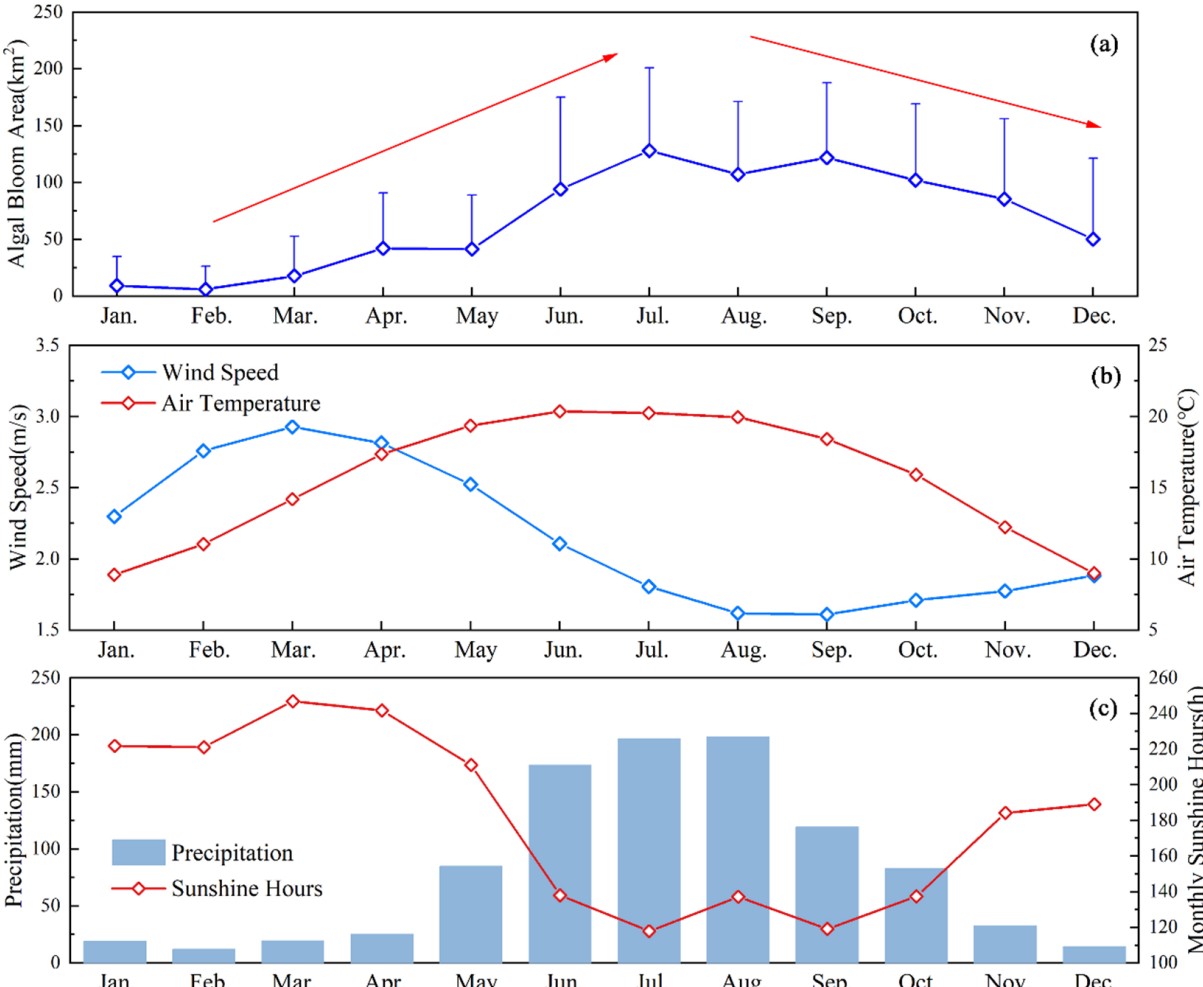

**Figure 5.** (**a**) Monthly algal bloom area in Lake Dianchi generated from all Landsat and MODIS/Terra images during 1987–2021. (**b**) Monthly wind speed and air temperature in Lake Dianchi. (**c**) Monthly precipitation and hours of sunshine in Lake Dianchi.

### 3.3. Spatial Distribution of Algal Blooms

Figure 6 shows the algal bloom frequency in each pixel (250 m × 250 m) in the whole lake. In terms of interannual changes, the bloom frequency in 2000, 2002, 2006, 2008, 2012, 2013, 2019, 2020, and 2021 was relatively high. Algal blooms in 2002, 2006, 2008, 2012, 2013, 2019, 2020, and 2021 had a high degree of intensity and a wide spatial distribution. From the perspective of the spatial pattern, before 2010, the bloom frequency in the northern region of the lake was significantly higher than that in other regions, showing an unmistakable pattern of a high bloom frequency in the north and a low bloom frequency in the south. For

many years, the bloom frequency in the northern region of the lake exceeded 50%. In 2001, 2005, and 2009, the bloom frequency in the northern region of the lake (>40%) was much higher than that in other regions (<20%), indicating that algal blooms basically occurred in the northern region of the lake in these years. After 2010, the difference in the bloom frequency between the northern region of the lake and other regions decreased, and the bloom frequency in the northern region of the lake decreased (to below 50% except in 2020), but the pattern of a high bloom frequency in the north and a low bloom frequency in the south did not change. The bloom frequency in the southern region of the lake increased, exceeding 20% in many regions. As for the average bloom frequency over the years, the northern region of the lake had the highest frequency (>40%), followed by the central region of the lake (20–30%). The southern region of the lake had the lowest frequency (<20%).

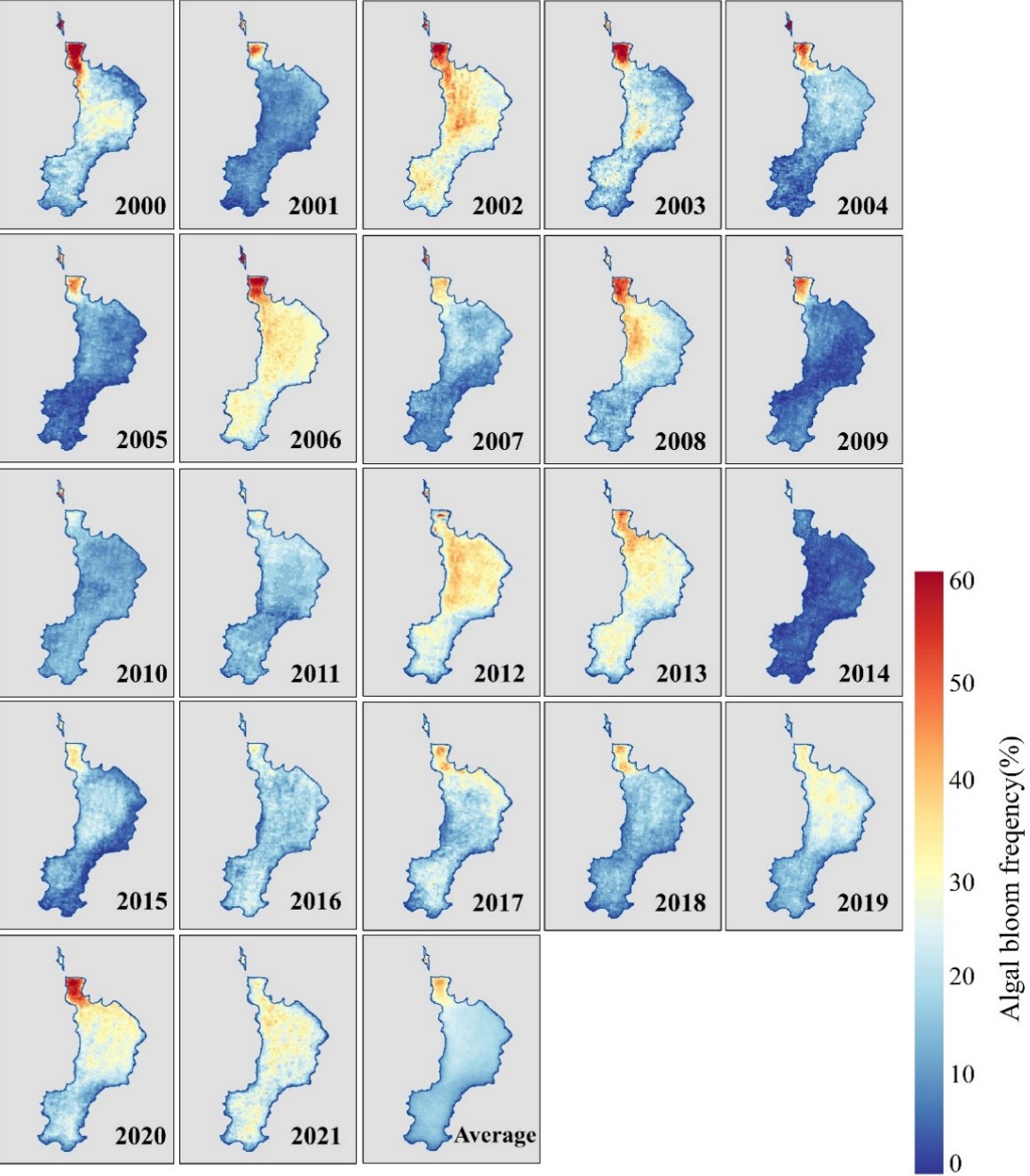

**Figure 6.** Annual bloom frequency distributions derived from MODIS/Terra imagery (2000−2021) for Lake Dianchi in China.

From the perspective of seasonal characteristics (Figure 7), the bloom frequency in the entire lake from January to February was low (January, 3.57% on average; February, 2.33% on average), and the bloom frequency in the entire lake was not higher than 20%. From March, the bloom frequency gradually increased (by 7.23% on average), the pattern of a high frequency in the north and a low frequency in the south gradually formed from April to May, and the bloom frequency in the northern region of the lake was close to 50% in May. Consistent with the results reported in Section 3.2, Lake Dianchi reached the peak period of algal blooms from June to November, algal blooms spread from the northern area of the lake to other areas, and the bloom frequency in the entire lake increased significantly. From July, the bloom frequency exceeded 60% in most regions in the northern area of the lake and was greater than 40% in the central area of the lake. The bloom frequency decreased from November to December, the bloom frequency in the high-bloom-frequency regions of the northern area of the lake decreased significantly, and the bloom frequency in the other lake regions decreased to less than 50%.

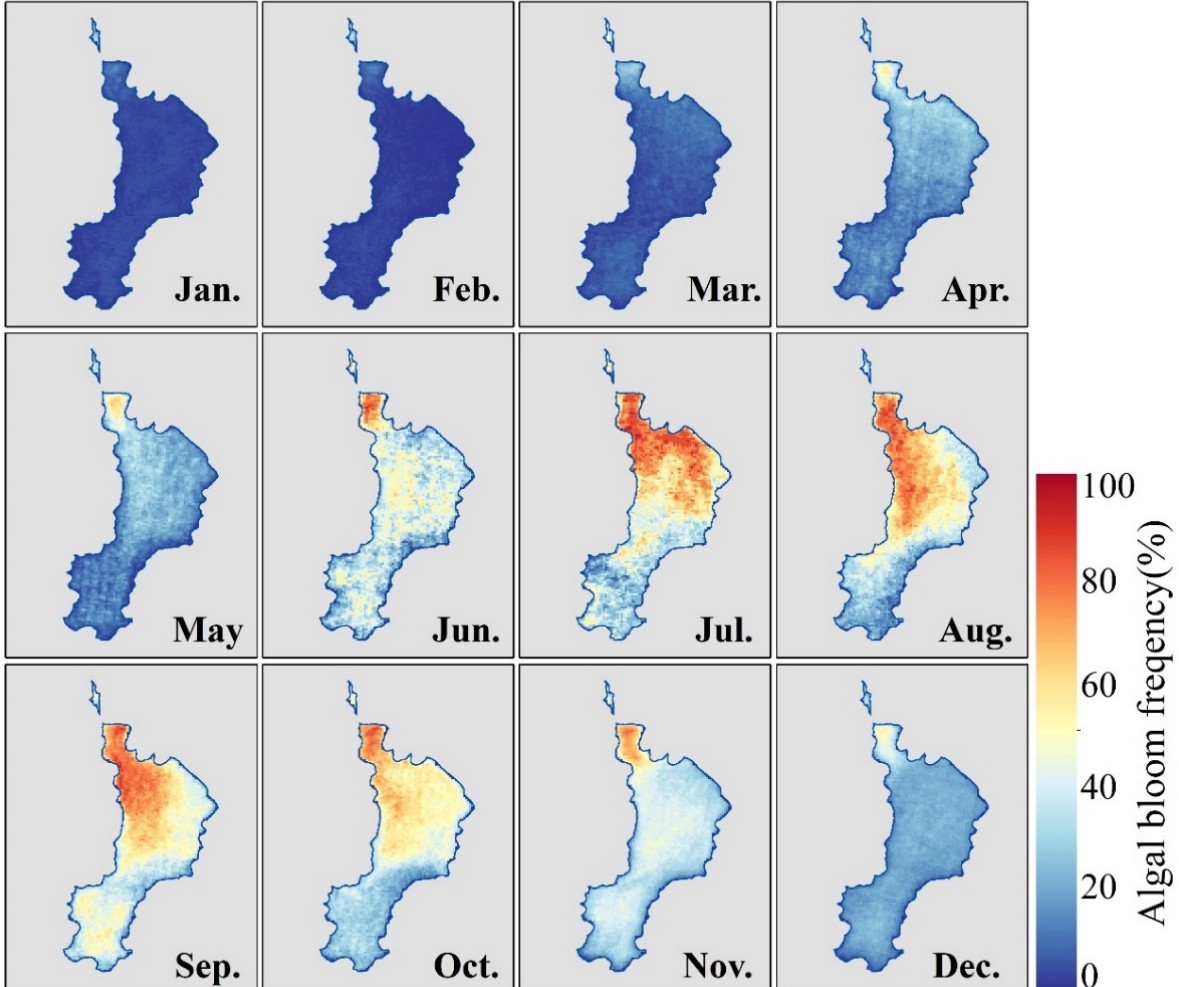

**Figure 7.** Monthly average bloom frequency distributions derived from MODIS/Terra imagery (2000−2021) for Lake Dianchi in China.

### 3.4. Temporal Characteristics of Algal Blooms

Figure 8 shows the change in the initial bloom time and the duration of algal blooms. From 2000 to 2021, the initial bloom time in Lake Dianchi showed a trend of first delaying and then advancing (Figure 8a). From 2000 to 2011, the initial bloom time tended to be delayed (y = 2.9895x − 5885, *p* = 0.0749). From 2000 to 2009, the initial bloom time fluctuated from day 76.66 to the 124th day, and in 2009–2011 it was delayed from the 108th

day to the 148th day. From 2011 to 2021, the initial bloom time was significantly earlier ($y = -7.8121x + 15858$, $p = 0.0034$). From 2011 to 2016, except for 2013 (on the 91st day), the initial bloom time in Lake Dianchi was later than the 100th day. After 2014, there was a significant decrease in the initial bloom time (from day 155.66 to the 76th day (in 2021)).

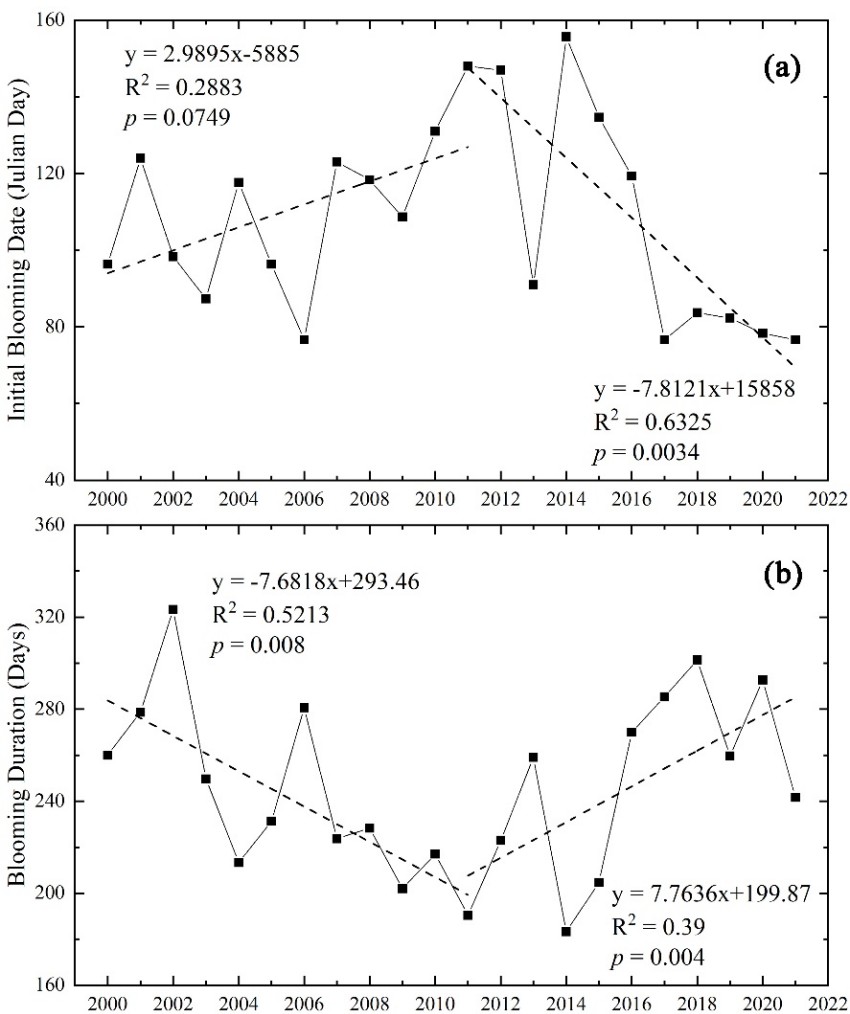

**Figure 8.** Trends in the initial bloom date and bloom duration for Lake Dianchi. (**a**) The initial bloom date throughout the time period of 2000–2021. (**b**) The bloom duration throughout the time period of 2000–2021.

From 2000 to 2021, the duration of algal blooms in Lake Dianchi showed a trend of first decreasing and then increasing (Figure 8b). From 2000 to 2011, the fluctuation in the duration of algal blooms decreased, showing a significant downward trend ($y = -7.6818x + 293.46$, $p < 0.01$). The duration increased from 260 days to 323 days in 2000–2002 and then decreased to 213 days in 2004. The duration increased to 280 days in 2006 and then decreased to 190 days in 2011. The duration of algal blooms from 2011 to 2021 showed a significant upward trend ($y = 7.7636x + 199.87$, $p < 0.01$). The duration increased to 259 days in 2011–2013 and then decreased to 183 days in 2014. From 2014, the duration continued to increase and remained at around 280 days from 2016 to 2021.

## 4. Discussion

### 4.1. Uncertainty in Long-Term Record Reconstruction Based on Multi-Source Satellite Data

#### 4.1.1. Observation Frequency of Multi-Source Satellites

The trends in the algal bloom time series reconstructed by Landsat and MODIS/Terra differed due to various factors (Figure 2b). Due to the characteristics of rapid changes in

algal blooms within a short period of time [36–39] and the significant seasonal distribution (e.g., the algal blooms in Lake Dianchi are concentrated into the rainy season), the difference in the observation frequency between Landsat (16 days/period of time) and MODIS/Terra (1 day/period of time) may lead to errors in the time series of algal blooms obtained within the same period. We calculated the relative error in the annual average bloom area obtained by Landsat and MODIS/Terra and the percentage of valid observations in the rainy season of the year. The results show (Figure 9) that the difference between the percentage of observations in the rainy season and the relative error had a significant negative correlation ($y = −4.2888x + 66.7497$, $p < 0.05$), i.e., the closer the percentage of images in the rainy season of the two sensors, the smaller the difference between the two time series. The algal bloom in Lake Dianchi had significant seasonal characteristics (Section 3.2), with a larger area and a higher frequency in the rainy season and a smaller area and a lower frequency in the dry season. When the percentages of observations in the rainy season from MODIS/Terra and Landsat are close, the obtained time series do not contain the errors caused by seasonal differences, resulting in a minor trend difference.

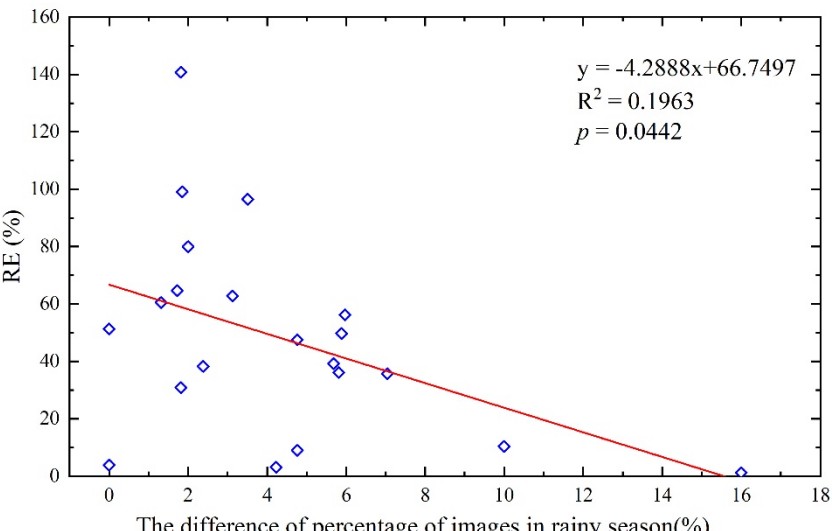

**Figure 9.** The relationship between the difference in the percentage of images in the rainy season and the relative error (RE) in the annual average area of algal blooms between Landsat and MODIS/Terra.

Differences in time series are also related to the time scales. The time series of the average bloom area obtained by Landsat and MODIS/Terra at different time scales were compared. Figure 4 shows the trend difference between the two time series with the change in the time scale. The results on the two-year scale show a minor difference in the trend and more detail, so the time scale for reconstructing the time series was determined to be two years. We compared the bloom areas extracted by Landsat and MODIS/Terra from 2000 to 2021 (Figure 10). The bloom areas on the two-year scale were closer to the annual bloom areas obtained by MODIS/Terra. The trends obtained from MODIS/Terra and Landsat are consistent, although there are differences in some details (e.g., extreme values) and the average bloom area decreases and then increases from 2000 to 2019. There are some differences between the Landsat time series and the MODIS/Terra time series in 2020–2021. We examined the original images and found that only two of the nine valid Landsat-8 OLI images from 2021 were obtained in May–October due to the rainy season, which is significantly different from the number and distribution of MODIS/Terra observations. None of the nine Landsat-8 OLI images contain significant algal blooms, while MODIS/Terra observed multiple significant blooms. This mismatch led to the discrepancy between the Landsat monitoring results and the MODIS/Terra monitoring results and is the reason why we wanted to obtain a unified time series by changing the observation interval.

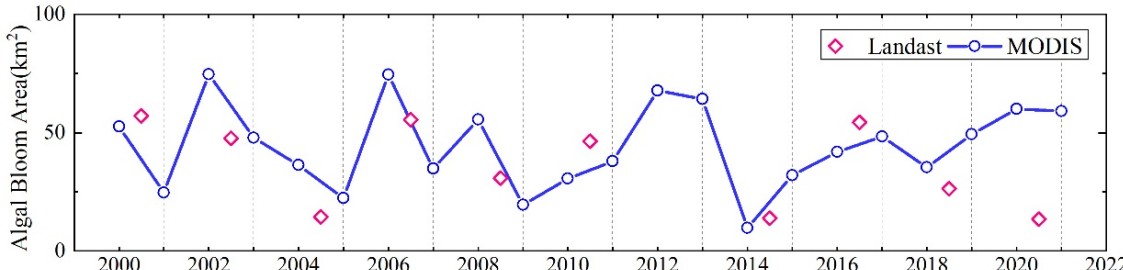

**Figure 10.** Comparison of the algal bloom area between Landsat (every 2 years) and MODIS/Terra from 2000 to 2021.

### 4.1.2. The Effect of Algal Bloom Indicators

We also tried to reconstruct the time series of the maximum bloom area from a combination of Landsat and MODIS/Terra images (Figure S4). The results show that as the time interval increased, the difference in the time trend between MODIS/Terra and Landsat became larger, indicating that it is difficult to establish a unified, comprehensive time series with images from many years. On the one hand, the average bloom area is more representative of the current period when combining data from different time scales. On the other hand, the key to the maximum bloom area in a certain period is whether effective observations are obtained on the day when the maximum bloom area occurs, so it has a higher degree of contingency due to its indicator definition. Therefore, the widening gap in the number of observations leads to a widening gap between the time series trends obtained by Landsat and MODIS/Terra.

### 4.1.3. The Effect of the Spatial Resolution of Multi-Source Satellites

Differences in spatial resolution may also lead to differences in the time series of algal blooms obtained with different sensors. Since high-resolution images may capture larger bloom areas due to the higher number of algal bloom features, low-resolution images may capture larger bloom areas due to the aggregation of more bloom features in mixed pixels [40]. The "truth" is often difficult to determine due to the resolution of different sensors [41], so the premise of reconstructing time series using different sensors is that the results obtained from different sensors need to be proven to have good consistency. In this study, Landsat (30 m) and MODIS/Terra (250 m) were used to reconstruct the time series of algal blooms, and 36 Landsat and MODIS/Terra images were obtained on the same day for comparison (Figure S5). The results show that the areas obtained by Landsat and MODIS/Terra are in good agreement ($R^2$ = 0.95). The difference in the bloom area between Landsat and MODIS/Terra ranged from 0.12 to 43.72 km$^2$, with 55.56% in the range of 0–10 km$^2$, 19.44% in the range of 10–20 km$^2$, 16.67% in the range of 20–30 km$^2$, and 8.33% in the range of 30–50 km$^2$. The most significant difference between Landsat and MODIS/Terra occurred on November 28, 2018 (MODIS/Terra, 100.87 km$^2$; Landsat, 57.14 km$^2$) due to the existence of thin clouds over the MODIS/Terra satellite, resulting in significant errors. In conclusion, the bloom area results of Landsat and MODIS/Terra can meet the requirements of time series reconstruction.

### 4.2. Effect of Environmental Factors on Algal Blooms in Lake Dianchi

The environmental factors that lead to the development of algal growth are complex, as they often involve the interaction of multiple environmental elements, such as meteorology and water quality [42]. These factors also have different effects on algae in different areas of the lake due to the geographical location and environmental characteristics of the lake itself. In order to explore the relationship between algal blooms and environmental factors over a long period of time and understand the process and mechanism of the occurrence and development of algal blooms in Lake Dianchi, we analyzed the relationship between the reconstructed time series of the bloom area and meteorological and water quality factors.

### 4.2.1. Meteorological Conditions

Wind speed is the key to algal bloom formation, and previous studies have generally concluded that lake surfaces require low wind speeds for algae to rise and form floating scums [43,44]. From 1987 to 2020 (Table 1), the monthly wind speed at Lake Dianchi varied from 0.8 to 4.4 m/s. In terms of interannual variation (Figure 4b), the average wind speed first decreased, then increased, and then remained stable. From 1987 to 1997, the average wind speed at Lake Dianchi decreased from 2.21 m/s to 1.61 m/s in 1997 and then increased to 2.8 m/s in 2006. Since then, the average wind speed at Lake Dianchi has remained at around 2.5 m/s. Air temperature is also a key factor affecting the growth of algae. A suitable air temperature will promote the growth of algae and aggravate algal blooms [42,45,46]. Since 1987 (Table 1), the monthly air temperature at Lake Dianchi has changed from 5.6 to 21.9 °C, showing a significant upward trend as a whole ($p < 0.05$) (Figure 4b). Precipitation will transport pollutants in the watershed to the lake, increasing the degree of eutrophication of the lake, which is conducive to the growth of algae [47–49]. Since 1987 (Table 1), the monthly precipitation at Lake Dianchi has changed from 0 to 474.9 mm. From 1987 to 2020 (Figure 4c), the annual precipitation at Lake Dianchi first increased, then decreased, and then increased again. Sunshine hours are also necessary factors for algal growth. A proper number of sunshine hours can promote algal growth [24,50], although some studies have shown that the high altitude (1887.4 m) of Lake Dianchi results in strong UV light that inhibits algal growth instead [47]. Since 1987 (Figure 4c), the monthly number of sunshine hours at Lake Dianchi has varied between 44.5 and 322 h. We compared the relationships between interannual meteorological factors and algal bloom parameters (Table 2). The results show that wind speed and air temperature are correlated with algal blooms. Specifically, wind speed was significantly negatively correlated with bloom duration time ($r = -0.5$, $p < 0.05$), and the average ($r = 0.36$, $p < 0.05$), maximum ($r = 0.38$, $p < 0.05$), and minimum ($r = 0.39$, $p < 0.05$) air temperatures were significantly and positively correlated with bloom area, indicating that the interannual trends of algal blooms in Lake Dianchi are mainly influenced by wind speed and air temperature. However, the correlations between precipitation, sunshine hours, air pressure, and algal bloom indicators are poor (Table 2), indicating that these factors are not the main driving factors of the interannual variation in algal blooms in Lake Dianchi [24]. We further selected for analysis the bloom areas from March to May, 2019 (Figure 11). There is a negative relationship between the bloom area and the wind speed on the day (Figure 11a,b); that is, when the bloom area increases, the wind speed on the day decreases compared with the previous day, and when the bloom area decreases, the wind speed on the day increases compared with the previous day. The relationships between air temperature, air pressure, and sunshine hours and the trend of the bloom area are not significant (Figure 11c–e).

**Table 2.** Relationship between the initial bloom date, bloom duration, and bloom area in Lake Dianchi and climate variables.

| | | WS | AT | ATmax | ATmin | PP | SH | AP |
|---|---|---|---|---|---|---|---|---|
| Initial Bloom Time | r | 0.39 | −0.21 | 0.09 | −0.14 | −0.11 | −0.19 | −0.27 |
| Duration of Bloom | r | −0.50 * | −0.01 | −0.25 | 0.04 | 0.36 | 0.03 | 0.25 |
| Bloom Area | r | 0.25 | 0.36 * | 0.38 * | 0.39 * | 0.03 | 0.26 | −0.28 |

* $p < 0.05$. WS, wind speed; AT, air temperature; ATmax, maximum air temperature; ATmin, minimum air temperature; PP, precipitation; SH, sunshine hours; AP, air pressure.

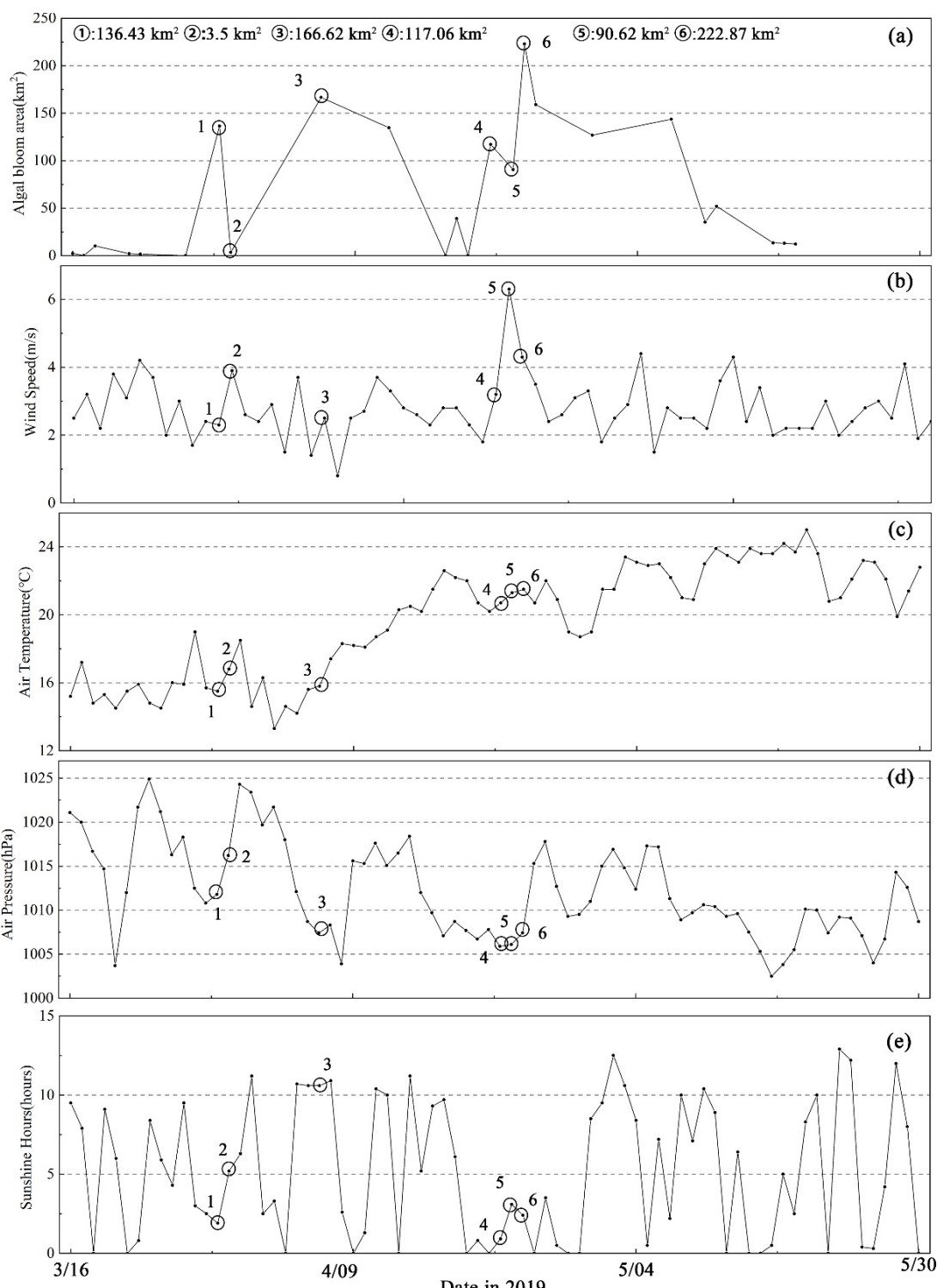

**Figure 11.** Algal bloom area, wind speed, air temperature, air pressure, and sunshine hours. The black solid circles in (**a**) represent the bloom areas derived from MODIS/Terra data. (**b**) Mean daily wind speed, (**c**) mean daily air temperature, (**d**) mean daily air pressure, and (**e**) mean daily sunshine hours. The numbered circles represent the 6 days during which the algal bloom areas changed greatly due to strong winds that triggered the dissipation of algal blooms on the water surface at Lake Dianchi.

Since the FAI was used to monitor algal blooms in Lake Dianchi, the algal blooms analyzed in this study were also defined as algal scums floating on the water surface [16],

which is also in line with the definition used in previous studies [28,51,52]. From the perspective of floating scums, the most direct effect of wind on floating scums is a reduction in the formation of floating scums due to mixing in the upper water-column, which is reflected in our long-term (Table 2) and short-term (Figure 11) monitoring results and is consistent with previous studies conducted in other regions [28,53]. Some studies have also found that vertical mixing due to high wind speeds can affect changes in the concentration of such substances as salt [54]. However, the question of whether similar phenomena can occur in shallow lakes such as Lake Dianchi (whose average depth is 5 m) remains to be further investigated. On the other hand, wind-induced waves may change the shape of the water surface, leading to phenomena such as sunglint, which may change the reflectance spectrum of the water surface. However, the FAI itself, due to its baseline subtraction design, is able to provide a more homogeneous background compared with indices such as the NDVI and the EVI in the face of a complex water surface (e.g., a surface subject to sunglint) [16]. Therefore, we used the FAI to extract algal blooms, and the reliability of the results was higher thanks to its relative stability under variable environmental and observational conditions. However, as temperature affects the rate of growth of algae and the intensity of lake stratification [8,47,54], the effect of temperature will be reflected to a greater extent by the long-term changes in algal blooms (Table 2).

Regarding seasonal variation (Figure 5b,c), there was a significant seasonal difference between meteorological factors at Lake Dianchi during the rainy season (May–October) and the dry season (November to the following April). The wind speed at Lake Dianchi was low during the rainy season (the average wind speed during the rainy season was 1.89 m/s, which is below the wind speed threshold of 3 m/s for algal bloom formation) [2,55] and the air temperature was high (the average air temperature during the rainy season was 19.04 °C, which is close to the cyanobacterial dominance threshold of 20 °C) [42], which is suitable for the rapid growth of algae. At the same time, the high precipitation at this time and the input of nutrients also promoted algal growth. These conditions led to a larger algal bloom area in Lake Dianchi during the rainy season compared with the dry season.

### 4.2.2. Nutrient Conditions

In general, the presence of an abundance of nutrients in the water column is a prerequisite for algal blooms, and numerous studies have shown that both nitrogen and phosphorus in the water column can be limiting factors for algal growth [56–60]. With the gradual acceleration of industrialization and urbanization in the phosphorus-rich mountainous regions of southwest China and the increase in the intensity of agricultural production, Lake Dianchi has become one of the most severely eutrophic lakes in China, and the high concentration of nutrients in the lake is an important reason for the continuous algal blooms [8,61]. The algal blooms in Lake Dianchi have attracted the attention of the local government. A series of lake management and pollution control measures have been implemented, and the concentration of nutrients in Lake Dianchi has been reduced [7]. Overall (Figure 4d), the TP in Lake Dianchi showed a significant downward trend ($p < 0.01$) from 1987 to 2020; specifically, an increase followed by a decrease. From 1987 to 2009, the TP showed a significant ($p < 0.05$) upward trend (TP in 1987, 0.20 mg/L; TP in 2009, 0.40 mg/L); after 2010, it showed a significant downward trend ($p < 0.01$); and, in 2019, it dropped to the lowest level (0.04 mg/L) of the entire period. From 1987 to 2020, the TN in Lake Dianchi also showed a decreasing trend (Figure 4d), specifically an increase followed by a decrease, but it was not significant (p > 0.05). From 1987 to 2009, the TN increased significantly ($p < 0.05$) (TN in 1988, 2.58 mg/L; TN in 2009, 5.06 mg/L), and, after 2010, the TN decreased significantly ($p < 0.01$) to the lowest level of 1.06 mg/L in 2019. In terms of the spatial pattern, most of the eastern and northern region of the lake is surrounded by the urban area of Kunming City, and many rivers flow through the city into the lake. These rivers discharged a large amount of urban and agricultural sewage into the lake, resulting in a significant excess of TN and TP, which is the reason for the higher frequency of algal blooms in the eastern and northern regions of Lake Dianchi.

The TN/TP ratio affects the composition of algal species, and a low TN/TP ratio favors the growth of cyanobacteria and promotes algal blooms [42,62]. During the entire observation period (Figure 4d), the TN/TP ratio showed a significant upward trend ($p < 0.01$). From 1987 to 2009, the TN/TP ratio at Lake Dianchi remained low (<15), and, after 2009, it increased significantly. Our results show that algal blooms in Lake Dianchi have not been significantly curbed in recent years (Figure 4a) and there is no significant correlation with the trend of a reduction in the nutrient concentration (Table 3). This is because, although the nutritional status of Lake Dianchi has improved significantly in recent years, the levels of TN and TP and the TN/TP ratio in Lake Dianchi have significantly exceeded the demand for algal growth for a long time [42] and are higher than those of typical eutrophic lakes in China (such as Lake Taihu and Lake Chaohu) [61,63]. However, the TN/TP ratio of Lake Dianchi exceeded the threshold of cyanobacterial dominance for temperate lakes (TN/TP ratio = 29) in 2020 [64]. If the nutrient concentration and TN/TP ratio trends continue to improve, the algal blooms in Lake Dianchi may be curbed in a real sense in the near future.

**Table 3.** Relationship between the initial bloom date, bloom duration, and bloom area in Lake Dianchi and water quality variables.

|  |  | WT | pH | NH$_3$-N | TN | TP | TN/TP |
|---|---|---|---|---|---|---|---|
| Initial Bloom Time | r | −0.07 | −0.12 | 0.04 | 0.22 | 0.06 | 0.05 |
| Duration of Bloom | r | 0.04 | −0.29 | −0.17 | −0.31 | −0.12 | 0.00 |
| Bloom Area | r | 0.32 | 0.19 | 0.21 | 0.16 | −0.02 | 0.39 |

WT, water temperature; NH3-N, ammonia nitrogen; TN, total nitrogen; TP, total phosphorus; TN/TP, TN/TP ratio.

## 5. Conclusions

Algal blooms have plagued Lake Dianchi for a long time. A long-term time series of algal blooms is crucial to understanding the changes in the lake's ecological environment and reducing the harms caused by algal blooms and is also a prerequisite for realizing algal bloom prediction, early warning systems, and prevention measures. In this study, we obtained a 34-year time series of the bloom area in Lake Dianchi from 1987 to 2021 by combining Landsat and MODIS/Terra images. A unified time series of bloom areas was constructed in order to analyze the spatiotemporal dynamics of algal blooms in Lake Dianchi over the years. The results show that the bloom area in Lake Dianchi had an overall upward trend from 1987 to 2021. The bloom area in the rainy season was significantly larger than that in the dry season. In terms of the spatial pattern, the frequency of algal blooms in the northern area of the lake was higher than that in other lake areas, showing a pattern of a high frequency in the north and a low frequency in the south.

The relationship between environmental factors and algal blooms was analyzed based on the reconstructed long-term records of algal blooms. The results show that wind speed and air temperature are the main meteorological factors controlling the interannual variation in algal blooms in Lake Dianchi. Since the requirements for algal growth have already been met, nutrients do not have a significant effect on the algal blooms in Lake Dianchi. This study provides the longest-term record of the spatiotemporal dynamics of algal blooms in Lake Dianchi to date, data support for the study of algal blooms in Lake Dianchi, and new ideas for research on and the management of inland freshwater lakes throughout the world.

**Supplementary Materials:** The following supporting information can be downloaded at: https://www.mdpi.com/article/10.3390/rs14164000/s1, Table S1: Validation of the result of cloud masking threshold proposed in this study; Figure S1: Statistics of endmember reflectance in different bands; Figure S2: Example of cloud removal with a cloud threshold of $R_{rc,2130}$ = 0.0246; Figure S3: Statistics of FAI thresholds for Landsat images and MODIS/Terra images; Figure S4: Time series on different time scales of the maximum bloom area in Lake Dianchi between Landsat and MODIS/Terra (during 2000–2021); Figure S5: Comparison of the algal bloom area derived from MODIS/Terra and Landsat imagery.

**Author Contributions:** Conceptualization, J.M. and F.H.; methodology, J.M.; software, J.M.; validation, J.M., F.H. and H.D.; formal analysis, J.M. and F.H.; investigation, T.Q., Z.S., M.S. and D.M.; data curation, J.M.; writing—original draft preparation, J.M.; writing—review and editing, F.H., J.L. and H.D.; visualization, J.M.; supervision, J.L.; funding acquisition, J.L., H.D. and Z.C. All authors have read and agreed to the published version of the manuscript.

**Funding:** This study was financially supported by the National Natural Science Foundation of China (41971309, 41971314, 42101378) and the Provincial Natural Science Foundation of Jiangsu, China (BK20210989).

**Data Availability Statement:** Not applicable.

**Acknowledgments:** This study was supported by the Lake Watershed Science Data Center (http://lake.geodata.cn, accessed on 30 December 2021), the National Earth System Science Data Center, and the 14th Five-year Network Security and Informatization Plan of the Chinese Academy of Sciences (Grant No. WX145XQ06-04) (http://data.niglas.ac.cn, accessed on 30 December 2021).

**Conflicts of Interest:** The authors declare no conflict of interest.

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
