# Peer review of "Thirty-Four-Year Record (1987–2021) of the Spatiotemporal Dynamics of Algal Blooms in Lake Dianchi from Multi-Source Remote Sensing Insights"

_remotesensing, doi:10.3390/rs14164000_

Round 1
Reviewer 1 Report
1. A brief summary.
This effort addresses the formation and study of long time series of information products based on the synthesis of Landsat and Terra data, that characterize algal blooms in Lake Dianchi. Using the obtained information product time series, the features of spatiotemporal algal bloom variability are studied and compared with meteorological data.
Undoubtedly, the paper covers the results of thorough and time-consuming processing of large amount of data. The paper’s structure is clear and logical. As the main contribution of the article the actual revealing of the spatiotemporal dynamics of algal blooms in the Lake can be named, which is reflected correctly in the title of the paper. The methodology used by the authors is interesting (however, there are comments on it).
2. General concept comments.
а) In the methodology proposed by the authors to obtain the key product of the paper (algal bloom time series), some features are not fully justified, namely:
-Lines 192-193 say, “The thresholds for all the images were statistically analyzed to ensure a consistent threshold over the whole period”. Then a decision to use the 2-sigma threshold is made. Taking into account that the maximum gradient method was applied to all the images (including those images that may not have had algal blooms), there are concerns that the chosen threshold will cause a large number of false positives when detecting algal blooms. Considering this, the authors are invited to justify the choice of such a threshold in more detail. As a result, it is this threshold that affects all the estimates obtained, including the blooming area and frequency. Could the obtained estimates be unreasonably high?
- In general, the methodology is aimed at determining and analyzing the area of blooms, and their intensity is not assessed. Although, the authors seem to have every data required for this. Here it is necessary either to justify the use of areas (why areas only?), or to enhance the study to the integration of blooming intensity within the detected areas. When integrating, the relevancy of the 2-sigma criterion, which is quite hard to be substantiated, will decrease.
b) Section 4, the discussion is interesting, however, some comments are to be made:
- Subsection 4.1 repeats the key points of the paper;
- «It is indicated that from March to May 2019, some blooming zones were additionally selected for analysis» (Fig. 11). What explains the choice of this particular short period for analysis? How representative is this period in relation to the entire data sample?
- The authors do not discuss the fact that the wind affects the water surface (wind waves) and can cause changes in spectral reflectance and FAI errors. Probably, the observed features of the supposed instant ecosystem’s response to the wind conditions can be associated with this phenomenon.
3. Review and comments.
- Note, that satellite data are not completely logically described in the text and in the figures (see, e.g. lines 23-24, Fig. 2b, etc.). The use of Landsat and MODIS satellite imagery is declared. Whereas Landsat is the name of the series of satellites, and MODIS is the name of spectroradiometers aboard AQUA and TERRA satellites. Recommend to call such data, for example, Landsat and Terra satellite images.
- It is not clear from lines 158-159, what the testing was, and on the basis of what criterion the methods proposed earlier were rejected.
- Line 165 and the text below cause a number of questions. Judging by the materials attached to the paper, the authors have detailed data on the accuracy of cloud masking. Instead of the words ‘WELL’ (line 166) and ‘MOST’ (lines 168 and 169), it is better to give specific accuracy metrics (for example, confusion matrix).
Thank you so much for your work!
Reviewer 2 Report
Line 179: Correct typo in comma.
Line 205-206: Why only one value for the coordinates?
Line 206: what is “m” after “20-20h”?
Line 225: the slope part is not clear.
Line 277: “deteriorate” means got bad, but you intend to say decreased, may be rephrase it?
Figure 3: Label figures (1 day, 1 month, 1 year, 2 year…….).
What is ym?
The slope part is not very clear, what is x? dif is difference is slope, right? Clarify.
Line 250: define n.
Line 396:define CZCS
Figure 9: Y axes labels and units :the wording is little confusing since it is not displaying %, rephrase it.
Line 439-442: …..more contingent due to……
Not very clear, please rephrase it.
Line 553-555: Correct the grammar.
Reviewer 3 Report
The authors of the article have done a lot of work on the study of long-term data on algae blooms in Lake Dianchi.
The article is replete with superfluous information, which in fact does not carry a strong semantic load.
When I analyzing the presented material a number of questions and comments arise both to the presentation of the material and to the essence of the material being presented.
1. Why is the MODIS level 2 satellite information not used for analysis, which contains information about the color index (the Gordon and Wang technique). To substantiate the need for this particular methodology about the color index according to the data of the MODIS level 1.
2. Why are these wavelengths chosen for to generate ‘truecolor’? Why is there no wavelength of 870 nm?
3. The article says that Landsat 5,7 and 8 data were analyzed. It also says that the Landsat-7 ETM+ images used in this study include only images up to 2003, however, Figure 2 shows a comparison of Landsat and Modis, and does not indicate which Landsat data compared.
Question regarding the statement that the main meteorological factors controlling the interannual variability of algae abundance are wind speed and air temperature. The temperature of course affects the intensity of flowering, but also the light regime, as well as the nutrients can not but affect the productivity of any algae. The temperature on the surface of the lake decreases as a result of mixing caused by increased wind. Then there is a deepening of the flowering layer of algae. The explanation of the patterns you described is not entirely obvious, because air temperature and illumination according to satellite data have a direct relationship. For a better understanding of the processes, I recommend reading the following publications:
Spatial distribution and interannual variability of cyanobacteria blooms on the North-Western shelf of the Black Sea in 1985-2019 from satellite data
AA Kubryakov, PN Lishaev, AA Aleskerova, SV Stanichny, AA Medvedeva, Harmful Algae 110, 102128
AAleskerova, A Kubryakov, S Stanichny, P Lishaev, A Mizyuk
Cyanobacteria Bloom in the Sea of Azov According to Landsat Data
IssledovanieZemliizkosmosa, 52-64
The wind speed and its direction can only affect the intensity of mixing of algae in the layers of the reservoir, that is, their spatial distribution. Wind is waves, and waves are interference in measuring the brightness of the sea. Perhaps, instead of the influence of wind speed on the measurement of flowering, we mean the influence of the wave (caused by the wind) on the atmospheric correction algorithms for calculating the concentrations of blooms. The authors of the article should take into account the error flags in MODIS products. For example, the TurbidWater flag corresponds to excitement. We ask the authors to clarify what they mean by the influence of wind or to describe a physical explanation of the influence of wind on blooms, possibly supported by references to similar works on this topic (the influence of wind on blooms directly).
What is Figure 3(a) presented for? If 2 satellite signals are compared anyway, why does the scale still display data from 1987? The caption also says that figures a), (b), (c), (d), (e) and (f) correspond to the time intervals: 1 day, 1 month, 1 year, 2 years, 3 years and 4 years, respectively. But Figure 3a does not show that it contains data for each day. The amount of data based on the comparison of figures a and b does not differ much, although there is daily variability in one figure, and monthly in the other. Please clarify what conclusion was drawn based on the analysis given in paragraph 3.1.
4. What is the sense of the phrase: "In different periods, the average area of algal blooms showed fluctuating development, and the characteristics of growth and decline were not noticeable" (line 278-280)
5. What is the meaning of averaging and getting 1 value per year for characteristics such as wind speed and temperature? Based on Figure 4 of the averaging data, despite the maximum number of sunny days in 2021 for the entire study period, this did not affect the algal bloom in any way, which is very strange and unlikely in principle.
6. According to what data (Modis or Landsat) and, accordingly, for what time period of averaging were obtained Figure 5 (a) MonthlyareaofalgalbloominLakeDianchi. (b) Monthly wind speed and air temperature in Lake Dianchi. (c) Monthly precipitation and sunshine hours in Lake Dianchi.
7. Figure 6 is the most informative in this work, but the signature contains an error – the analyzed time period is incorrectly specified – not 2020-2021, but 2000-2021.
8. (line 404-407) Results showed that the difference between the percentage of observations in the rainy season and the relative error showed a significant negative correlation (y = -4.2888x+66.7497, p < 0.05), i.e., the closer the percentage of images in the rainy season of the two sensors, the smaller the difference of two time series. It is necessary to explain and correct the program of the proposal.
9. Figure 10 requires explanation. For example, as can be seen from the presented data on the area of algae blooming, in 2016 there is a difference of satellite measurements by 2 times, and for data for this year a large area of flowering is observed according to Landsat data, and for 2021 the situation is diametrically opposite. The authors need to explain the situations of data differences for each case when the difference is more than 25%.
10. Paragraph 4.3.1 Meteorologicalconditions472 line. What is the reason for choosing 2019 as the analyzed year and the March – May season for this year.
I really hope that these comments and their elimination will help the authors to improve their work in general
Best regards
Round 2
Reviewer 1 Report
Dear authors!
Thank you very much for the prompt response to comments about your solid manuscript. Most of the comments were resolved (reflected both in the cover letter and in the updated version of the manuscript). However, two major comments (they will be further given below) received a response in the cover letter, but were not properly reflected in the updated version of the manuscript.
These comments are meant:
“
Comment 1: “Lines 192-193 say, “The thresholds for all the images were statistically analyzed to ensure a consistent threshold over the whole period”. Then a decision to use the 2-sigma threshold is made. Taking into account that the maximum gradient method was applied to all the images (including those images that may not have had algal blooms), there are concerns that the chosen threshold will cause a large number of false positives when detecting algal blooms. Considering this, the authors are invited to justify the choice of such a threshold in more detail. As a result, it is this threshold that affects all the estimates obtained, including the blooming area and frequency. Could the obtained estimates be unreasonably high?”
Response: Thank you very much for your comment. First of all, we apologize for the lack of clarity in the description of the threshold acquisition, which led to the misunderstanding of the reviewers. We used the maximum gradient method to determine the algal bloom extraction threshold from the practice of Hu, et al. [1] in Lake Taihu. In fact, we used FAI>0.02 and FAI<-0.01 to exclude pixels associated with thick algae scums and pixels associated with pure water and high concentrations of submersed particles (either algae or sediments), respectively [1]. The pixels that were retained were between pure water and thick algae scums, and at the edge of the algal bloom range. If a scene has no algae at all, then all pixels in the scene are excluded by FAI<-0.01, and no response threshold is generated to influence the final threshold determination.
In addition, after obtaining the response thresholds for the images with algal blooms considered for each imagery, histogram statistics were performed and the results showed an approximately normal distribution. The thresholds obtained by subtracting the mean value from the 2-sigma standard deviation were able to cover 95% of the images and avoid the false positives caused by the use of the minimum threshold in the statistics. The method has been commonly applied to studies of algal blooms, wetlands, water bodies, and mining vessels [1-3].
In summary, the method in this study excluded non-bloom pixels and a universal threshold for most images were obtained based on the statistical results of the approximate normal distribution.
Comment 2: “ In general, the methodology is aimed at determining and analyzing the area of blooms, and their intensity is not assessed. Although, the authors seem to have every data required for this. Here it is necessary either to justify the use of areas (why areas only?), or to enhance the study to the integration of blooming intensity within the detected areas. When integrating, the relevancy of the 2-sigma criterion, which is quite hard to be substantiated, will decrease.”
Response: Thank you very much for your comment. In this study, the bloom area rather than the bloom intensity was applied as an indicator to reveal the spatial and temporal variation of algal blooms in Lake Dianchi, because the bloom area made the only verifiable data we could use. Some studies did use bloom intensity as an evaluation indicator of algal blooms. Ho, et al. [4] used single band (near-infrared band) to characterize bloom intensity and applied it to the assessment of global lakes, but this work has been questioned by Feng, et al. [5] and the application of this method is doubtful. Binding, et al. [6] used pigment concentration as a measure of bloom intensity. Wang, et al. [7] modeled AFAI with measured biomass data and revealed the bloom intensity of Sargassum. However, all of their works required a large amount of simultaneous satellite and in-situ data. Although this study obtained water quality data from automated stations and filed surveys in Lake Dianchi, it lacked simultaneous data on biomass or pigment concentrations. Therefore, modelling and validation data on the bloom intensity are difficult to obtain in this study. From another perspective, the relationship between spectral indices (e.g. FAI and AFAI) or pigment concentrations of algal blooms and the bloom intensity is still unclear. The relationship between AFAI and algae biomass is not stable [7]. Moreover, the pigment concentration of the algal bloom area is difficult to be accurately obtained by remote sensing [5]. It can be seen that the current methods to obtain the intensity of algal bloom from algal bloom pixels are still immature and not the scope of this study.
In conclusion, this study chose the bloom area, which can visually reflect the severity of algal bloom, as an indicator to evaluate the spatial and temporal variations of algal blooms in Lake Dianchi.
“
It would be useful to include directly in the text of the manuscript the key points of the discussions given here on the justification of the 2-sigma criterion and the expediency/inexpediency of choosing a strategy for studying the area/intensity of blooms. This could give more clarity to the readers of the article and increase its scientific significance.
With best regards!
Reviewer 3 Report
Dear Colleagues!
I am very glad that the suggestions I made to improve the version of your work were received so positively. A lot of work has been done and, of course, all the corrections and explanations are made with high quality and absolutely meet all the requirements. Thanks for the work you've done! I will be glad to further cooperation!
All the best
